# A Qualitative Inquiry of a Three-Month Virtual Practicum Program on Youth with Visual Impairments and Their Coaches

**DOI:** 10.3390/ijerph19020841

**Published:** 2022-01-12

**Authors:** Lauren J. Lieberman, Lindsay Ball, Pamela Beach, Melanie Perreault

**Affiliations:** 1Department of Kinesiology, Sport Studies and Physical Education, State University of New York (SUNY), Brockport, NY 14420, USA; pbeach@brockport.edu (P.B.); mperreault@brockport.edu (M.P.); 2Department of Human Movement Sciences, Old Dominion University, Norfolk, VA 23508, USA; lball0006@odu.edu

**Keywords:** physical activity, pedagogy, blind, phenomenology, professional preparation

## Abstract

Research has shown that the practicum experience for professional preparation students in physical education teacher education programs related to teaching youth with disabilities can improve self-efficacy. It is not currently known if a virtual program can be effective for the professional preparation students or the participants. The objective of this study was to determine the experiences of the participants of a three-month virtual practicum program. In this phenomenological study, thirty youth with visual impairments and 1:1 professional preparation students (coaches) took part in a three-month virtual physical activity program. A total of 11 coaches took part in 2 focus groups, and 10 of the participants were interviewed about their experiences in this unique practicum. Findings in this three-month program revealed four themes: (1) friendship, (2) self-determination, (3) goal setting, and (4) barriers. The results of the qualitative inquiry indicate that a virtual practicum program can have a positive effect on both the participants and the professional preparation students. Virtual programs should also be aware of barriers to implementing an effective program to benefit all parties.

## 1. Introduction

Youth with visual impairments are often delayed in comparison to their same-age peers in physical activity levels and development of their motor skills and balance [1,2,3,4,5]. This developmental delay is often linked to fewer opportunities in their day-to-day lives to engage in physical activity and develop their motor skills [6,7,8]. In addition, youth with visual impairments are often left out of important decisions related to physical activity and health [9,10]. Though self-determination is one of the nine areas outlined in the Expanded Core Curriculum [11], many students with visual impairments do not have goals on their Individualized Education Program related to self-determination [12]. Unfortunately, this may contribute to students’ lack of self-determination, as they are not always provided opportunities to self-advocate or make choices for themselves. This combination of deficits often leaves youth with visual impairments with limited life skills. The confidence to change their lives as self-determination is imperative to one’s sense of self and well-being [9,13].

In addition to the physical activity and motor development deficits found in youth with visual impairments, their physical education teachers are often unsure what to teach them and how to include them in class safely. Teachers have expressed that students who have a visual impairment are one of the most difficult populations to teach in physical education [14]. In addition, individuals with visual impairments have reported that in physical education, they experienced greater differences between themselves and their peers [15], and their teachers did not know how to teach them basic skills, such as running [16]. These teachers feel that they never learned how to teach youth with visual impairments in their professional preparation programs [14]. To compound this fact, the multidisciplinary team members who could help physical educators the most are Teachers of Students with Visual Impairments (TVI) and Certified Orientation and Mobility Specialists (COMS). A recent study of 68 teachers in the field of visual impairment illuminated the fact that they also did not receive training in their professional preparation programs [17]. This lack of training often leads to reduced opportunities for youth with visual impairments in physical education activities.

Several studies have analyzed the effect of practicum programs on self-confidence in professional preparation students to teach youth with disabilities in physical education. Jeong and colleagues [18] compared the self-confidence of pre-service physical education teachers in teaching students with disabilities before and after the completion of their adapted physical education practicum. Results indicated that the self-confidence of the pre-service physical education teachers in working with individuals with disabilities increased, and 92% of the pre-service teachers found the practicum to be greatly beneficial or beneficial. A recent study also found that disability sports programs have shown gains in self-efficacy in pre-service physical education teachers [19]. Lastly, Hodge and Jansma [20] found that both off- and on-campus practicums promoted positive attitude changes in pre-service teachers. They also found that on-campus practicum experiences improved attitudes significantly more than off-campus ones.

Qualitative research of practicum experiences has revealed positive outcomes. Hodge et al. [21] conducted a phenomenological qualitative study to explore the meaning of the practicum program for physical education teacher education (PETE) students. Through the 8-week practicum experience with children with a wide variety of disabilities and weekly journaling, the authors found 11 themes that support the practicum. The main finding established that students’ attitudes and perceived competence were favorably influenced by what they viewed as challenging, rewarding, and meaningful practicum experiences. In another qualitative study, Standal and colleagues [22] conducted a yearlong program in a variety of settings both on and off-campus with 19 professional preparation students using field notes, journals, and focus groups. They found that issues concerning teaching methods and understanding of disability were important for the students. They also suggest that there are several other ways that students need to be supported to make sense of practicum experiences, such as discussions, support, and meetings. Lastly, Sato and Haegele [23] conducted a qualitative analysis on graduate students’ experiences instructing students with severe and profound disabilities. In this study, 9 participants who were enrolled in a practicum course worked with children with severe and profound disabilities for 200 h. Each student was interviewed, followed up through email correspondence, and filled out demographic questionnaires. Three themes that emerged through the use of organizational socialization theory were: (1) the ambiguous roles of APE teachers; (2) the need for specialized expertise; and (3) reality shock-unpredictable behaviors. Although each participant expressed some difficulties during the program, they also shared degrees of success in engaging in and contributing to the education of students with severe and profound disabilities.

Practicum experiences have also been shown to benefit youth with visual impairments. Lieberman and colleagues [24] studied the effect of a one-week sports camp on engagement in the nine components of the Expanded Core Curriculum (ECC), including self-determination and socialization. They found that a one-week educational sports and recreation program promoted engagement in all nine areas of the ECC multiple times and in multiple settings in the camp. In addition, a study by Brian and colleagues [6] investigated how perceived motor competence of youth with visual impairments improved after a one-week face-to-face sports camp. They found that after one week of instruction in sports, recreation, and self-advocacy, the children’s perceived motor competence improved significantly. Moreover, Shapiro et al. [25] determined that a one-week face-to-face educational sports camp improved self-worth for youth with visual impairments.

Despite the benefits for both pre-service teachers and youth with visual impairments, Layne and Balsingame [26] determined that physical education teacher education programs are challenged to find adequate time for field experiences. Moreover, the challenges of the COVID-19 pandemic have limited in-person opportunities for pre-service teachers to gain experience working with individuals with visual impairments. Thus, it is important to examine alternative practicum formats that could be used in physical education teacher education programs. Therefore, the purpose of this study was to examine the experiences of youth with visual impairments and their 1:1 pre-service teachers who participated in a three-month virtual practicum physical activity program. To examine these experiences, this study adopted an interpretive phenomenological analysis (IPA) approach to navigate data collection, analysis, and interpretation [27,28]. The qualitative approach of IPA examines how participants view their personal and social worlds and the meaning their lived experiences within those worlds holds for them [28]. Understanding the physical activity practicum experiences of pre-service teachers and youth with visual impairments is imperative to strengthen the effectiveness of this educational opportunity.

## 2. Materials and Methods

### 2.1. Participants 

The Institutional Review Board of the lead researchers’ institution approved this study. A call for participants ages 6–19 years with visual impairments went out on listservs and social media groups. Thirty youth with visual impairments volunteered to engage in a three-month virtual practicum. The 1:1 coaches included pre-service teachers in the field of physical education teacher education who were involved in one of two adapted physical education classes. An email went out to families who signed up for the program to participate in the qualitative portion of the study. In the end, 10 youth with visual impairments (athletes) from New York, Massachusetts, Georgia, and California signed up for the study. See Table 1. for the demographic data of the participants. Pseudonyms were used for all athletes and for coaches who preferred one. In addition, the pre-service teachers (coaches) in two adapted physical education classes were invited to participate in two different focus groups. Eleven of the coaches agreed to participate. See Table 2. for the demographics of the coaches who were interviewed. Lastly, the practicum instructor and one teacher from a school for the blind where several participants went to school were interviewed about their reflections of the practicum program. 

### 2.2. Virtual Practicum Description

The overall goal of the program was for the athletes to improve their motor skills and physical activity and for the coaches (pre-service teachers) to learn how to teach youth with visual impairments. The virtual practicum took place over three months. First, athletes were paired with undergraduate pre-service teachers (coaches) in a physical education teacher education program in the northeast of the United States and provided with each other’s contact information. The practicum required that the pairs meet a minimum of once a week for 60 minutes and work on fitness as part of the program; however, they could choose the virtual platform, when to meet, how often, and what physical activities to work on. The program started with an opening ceremony attended by the athletes, their parents, and their 1:1 coach and included an explanation of the virtual practicum. At this time, each athlete also completed a short fitness assessment consisting of four components: duration of a wall sit, number of push-ups, number of sit-ups, and duration of a plank. Six weeks into the program, there was a “check-in” where the athletes could share what they had been doing and what they were proud of. Additionally, the four fitness components were re-assessed to determine improvement. This “check-in” included elite athletes sharing their stories of their journey into elite sports. The last week of the program consisted of a closing ceremony where every athlete completed their final fitness assessment and shared their accomplishments. There was also an elite athlete sharing his accomplishments and providing words of wisdom for success in the future. 

### 2.3. Interviews and Focus Groups

This qualitative phenomenological study included semi-structured interviews and focus groups related to the three-month virtual practicum. Phenomenological research studies examine the essence of the lived experience of an individual or a group of individuals related to a specific phenomenon [29]. The phenomenon that was analyzed in the current study was the experiences of youth with visual impairments and their 1:1 coaches during a three-month virtual practicum for two adapted physical education courses.

Semi-structured interviews and focus group scripts were developed by the lead researchers to ensure the totality of the lived experience of the participants and their coaches was captured in the data. The lead researchers began with the end in mind by developing questions that would provide the answers to critical information needed to understand the effectiveness of the program. The overall purpose of the questions was to understand the lived experience of the virtual program by the athletes and the coaches. The scripts were sent to six experts in the field to ensure face and content validity. There was one athlete who is blind who is also a physical education teacher, three professors in motor development and adapted physical education who have expertise in working with youth with visual impairments, one parent, and one teacher of youth with visual impairments. Some of the guiding questions for the focus groups were as follows: (1) How did you measure improvement? (2) What was motivating to your child? (3) What about the personal best program was motivating to you? (4) Did you find this program enjoyable? Why or why not? (5) Were there any barriers to executing your lessons? (6) What did you find helpful to completing your lessons? (7) How did you feel about the online format of the program? (8) What could make this program better? The guiding questions for the athlete interviews were as follows: (1) How would you summarize your experience in the virtual program? (2) Do you feel that you made any new friends or that you became closer to any of the other athletes during the course of the program? (3) What barriers to being involved in physical activity did you experience during this program? (4) What helped you engage in sports and physical activity during this program? (5) As a result of your involvement in this program, what changes did you experience in yourself? (6) How do you feel this program helped you self-advocate for what you need in sports and recreation? (7) As a result of your involvement in this program, what improvements did you notice in your locomotor and ball skills? (8) As a result of being involved in this program, how do you feel you can be more involved in sports and recreation in your school and/or community? (9) How could this program have been better for you?

The researchers conducted online interviews using the scripts with the athletes at the end of the three-month program to capture their experience during the virtual practicum. Each interview involved asking a series of structured questions and subsequently probing more deeply with open-form questions to acquire additional information [30]. At the end of camp, the investigators also conducted two in-person focus groups with the athletes’ 1:1 coaches. The goal was to better understand the coaches’ experiences during the three-month practicum [31]. The focus groups consisted of seven and four coaches, respectively. Lastly, the practicum instructor and teacher at the school for the blind were interviewed about a month after the program ended. All interviews and focus group discussions were audiotaped to confirm that the researchers documented the substance of the experiences and could come to an agreement on the results. The transcribed interviews and focus group discussions were then transcribed verbatim.

### 2.4. Data Analysis

Three of the qualitative researchers analyzed the interview and focus group transcriptions. The investigators first independently coded the transcripts for major themes, subthemes, and supporting quotes to fully understand the entirety of the transcript data [32]. The researchers used Braun and Clarke’s [33] advice on completing the thematic analyses to ensure they managed coding the data in a theoretically and systematically sound way. After the initial coding, the three researchers then met to assess common codes, discuss variations of codes, and review emerging themes. In the discussions, the themes and supporting sub-themes developed and were discussed with continuous analysis of the data, returning to examine it several times [34]. In the following meetings, the major themes and subthemes were reduced based on similarity of meaning and content. The themes and sub-themes used in this manuscript were agreed upon by all three researchers. 

## 3. Trustworthiness

The concept of trustworthiness refers to the level of confidence in the data, interpretation, and methods employed to ensure the quality of truth of a study [35]. Trustworthiness in this study was determined in several ways. The investigators (a) interviewed the athletes with validated research questions, (b) conducted two focus groups with the athletes’ 1:1 coaches with validated questions, and (c) included the practicum instructor and an involved physical educator to ensure triangulation of the data. Secondly, the ten athletes who were interviewed were randomly selected from the purposeful sample as described earlier. Each athlete was given the right to refuse participation in the study [36]. Lastly, the third researcher, a “critical friend” who was not involved in the virtual practicum, was on the research team to ensure that the themes and subthemes utilized were accurate and reflected the lived experiences of the athletes and their coaches. Because two of the researchers conducting the study were involved in the virtual practicum, and they knew the athletes and coaches, the unbiased and outside viewpoint of the additional investigator familiar with the field of study was necessary [37]. Therefore, the “critical friend” held the researchers accountable and confirmed an unbiased lens.

## 4. Results & Discussion

In this study, 30 youth with visual impairments engaged in a 3-month virtual practicum with 1:1 coaches who were pre-service teachers in the field of physical education teacher education. Each youth and their coach met at least once a week and chose their specific activities, virtual platform, meeting time(s), and the duration of each activity. The results highlight the effects of the program on both the athletes’ learning of skills and improved physical activity and the coaches’ understanding of pedagogical instructional approaches. Please note that these themes and sub-themes in the findings reflect the experiences of both the athletes and the coaches as they worked together and acquired these experiences together. Clarification was made in each quote if the response was from an athlete or a coach. The findings in this three-month program revealed three themes: (1) friendship, (2) self-determination, and (3) barriers.

### 4.1. Friendship: “We Just Kind of Formed a Bond”

Many of the athletes and coaches in the virtual practicum emphasized the value of forming friendships with one another over the course of the program.

Coach Heather shared:

We had something in common. We both play the same sport, which is wrestling, and also at the time, we’re both huge Marvel fans and at the time, we were watching two of the shows. It was just kind of like fun to like talk to him, and there’s just like there’s so much in common.

Coach Jamal stated that: 

…it was like it was kind of more than like a camp, it felt like I was just like talking to one of my friends, and we just kind of formed a bond, and I really appreciate it too. I’ll definitely keep talking to him for a long time.

Coach Jamal also noted: “Yeah in the end I loved it. I mean I made a friend out of it. So if anything else that was good.”

Staying connected every week was huge for so many of the participants. Coach Rachel shared, “She loves having conversations, she’s always like so excited to see us and just like having like that kind of a bond with her was like, made it worth it.” Brian, an athlete in the program, expressed this sentiment in his statement, “…with my coach, I definitely feel like we had a pretty good relationship.” Even the practicum instructor and the physical education teacher of several athletes observed the friendships form and noted the importance of the connection. 

The practicum instructor stated:

… just seeing the relationships that they built it was, it made me so happy to know that, like the kids aren’t as isolated as they were during COVID-19 because now they have this coach that cares about them and is meeting with them at least once a week.

She also shared: 

The concept of friendship and socialization was huge, and a lot of them considered their college coach their friend. And some of them were like, they’re my best friend, you know, because they’re meeting with them a couple times a week.

The physical education teacher had similar observations. She stated:

When I think of a few students in particular, I think that who they were paired with, they were really connected to them. I know a lot of my students looked up to who they were with and were excited to like hear from them and everything. 

She also noted, “One student he um, his coach invited him to watch her play lacrosse on zoom or something. Like her game was aired and I thought that was really cool.” It appeared that during this time of isolation, all of the participants enjoyed getting to know someone new. 

### 4.2. Self-Determination: “I Want Him to Have an Opinion” 

Another theme that stood out with most of the athletes was the notion of self-determination. Within this theme, three subthemes were evident: autonomy, competence, and self-advocacy.


**Autonomy: “Each week we’d take her feedback and switch up our own plans a little bit.”**


Many of the coaches in the program wanted to provide their athletes with a sense of autonomy by allowing them to be involved in the planning of the workout sessions. The coaches provided the athletes with opportunities to make choices about the exercises they did, the times they met, how they met, and the music they listened to. Coach Jamal expressed:

Typically, with Adam I would generate the activities that we were going to do. I like to give him choices too. I want him to have an opinion because if I’m doing something that he doesn’t enjoy as much, I’ll kind of veer away from that and try to go into other options for him. 

Along the same lines, Coach Heather stated: 

…we kind of would give her the choice because it was, we wanted to make it fun, it was a Thursday and she was almost done with the week, so she was just, it was a lot for her after school.

Coach Heather also shared:

I got his feedback on a couple but he said that he thought sit-ups did more than crunches for him so we would change something like that. Or he’d tell me, I really enjoy doing push-ups, so we added a little more push-ups. 

This idea that the coach started the line of ideas, and the athlete chose their preferred activities was a common finding. Coach Rachel said: 

…we’re like what activities do you want to do for this song, and she’s like, oh let’s do bicycle kicks and squats like she’s like, super involved with her own planning now, which is really cool. 

The concept was confirmed even more with Coach Jordan, who shared: 

With Tim, I generated the activities, but as he got stronger, he started adding to the activities. So, for example, if we were doing a jump squat, he would add a jump squat with a spin, a full spin. And he kind of did that pretty much to all of the exercises. Once he became stronger. But in the beginning, I generated it, and he kind of took over. Like even now, he’s taken over, he’s doing it by himself, really. And I have to keep up with him.


**Competence: “I can go longer, I can push myself harder”**


During this program, the coaches developed lessons and exercise plans that focused on fitness. Coach Heather expressed, “… we did a lot of just like exercise plans. We created about three or four together that work on just conditioning and different kinds of muscular strength and endurance workouts.” Coach Jordan stated, “We did a lot of muscular strength exercises and flexibility and core strength. A lot of crossbody type of strength for his balance as well. That was the main focus.” Nora reiterated this with her statement, “Just like exercises that you don’t need to have equipment for like lunges, wall sits, sit ups, high knees, jumping jacks, burpees, push ups, planks, squats, and things like that.”

Many of the athletes felt as though they achieved better fitness over the course of the 12 weeks. Brian enthusiastically said, “I noticed that with some of the exercises that I did, I would get, I would like say like get stronger in my arms or legs.” Coach Rachel also supported this notion:

I mean that like first field day that we had with everybody, she didn’t really know how to do a push up or like sit ups or anything like that. And now, I mean now she can go like a half hour, moving without really taking much of a break…

Coach Gary shared a similar experience:

I could tell that they’re doing more reps and holding planks longer, doing mountain climbers longer, running in place longer. My kid specifically was doing more push-ups, doing more burpees at a time. So, their cardiovascular endurance was increasing, and their muscular strength was increasing. 

Lastly, Adam expressed, “My endurance, my athleticism has changed. I can go longer, I can push myself harder, I can do more intense workouts than I could on the first day.”

Goal setting was a key component for increasing fitness competence throughout the practicum. This was part of the philosophy of the program and a mantra that the participants followed. Brian clearly shared this when he said, “I thought it was pretty cool it was something different, because you could set your own goals for different physical activities that you would want to do….it was all about meeting goals.” Another participant, Adam, highlights the importance of goal setting for tracking progress:

I’m almost there to reaching my goal that I had for this program. My goal for this program was to have both of my skate blades, touching the pipes on the net. Six feet across 180 degrees out. And I’m almost there. I’m like so close, probably by next Saturday. I’m going to have it.

This was also expressed by another coach:

I mean I think we met our goal with Latisha being able to run and stuff, like she knows how to move her arms, she knows to move her legs and arms at the same time. And, Emma and I both said it looked good. She came a long ways from where she was.

Coach Jordan highlighted the motivational side of goal setting for both coach and athlete. He stated:

For Tim, it was just seeing him light up every time I would log on to zoom and he was ready, he had a smile on his face. Every time we started, he would complete the task he was ready to go right to the next exercise. And that just motivated me to, you know, cuz he wanted to do it. I wanted to make sure that I gave him, you know, the very best that I could to help him to reach his goals.

The physical education teacher also agreed with this concept, saying, “I think overall their goals were pretty much met, and they kept consistent, and they were motivated because they kind of had set goals for themselves before we started.”


**Self-advocacy: “It helped me self-advocate”**


The athletes described how the program gave them the tools and language to self-advocate for themselves during and outside of the program. For example, Adam stated:

It helped me self-advocate, because some of the exercises that I didn’t, that were unfamiliar to me I would ask the coaches to explain it, which is a big part of self-advocacy asking someone to verbally explain what they’re doing, because I’m a B1 I cannot see what the motions they’re doing on my iPhone.

Latisha’s mother offered a similar example. She shared: 

… in the middle, she [Latisha] just like started to begin directing the session where she would try to convince us like no I’m gonna, I’m gonna do sit ups instead of jumping jacks right now. And then is like, well, you want to make me do sit ups, give me an object on my feet so I can come up and get it and lay back again.

The practicum instructor also supported this notion, stating, “So, he was very clear on what he wanted. How often he could meet, what his goals were. And the more that the child knew what they wanted and could communicate the better the situation was.” The physical education teacher reiterated the sentiment of self-advocacy. She shared: 

I had a high schooler who participated, and she’s like how do I let them know like what’s going on with me and I, so we kinda talked about how you can share with other people, what you can say, like its ok to say I don’t know what you’re talking about, can you explain that in a different way.

Even the youngest athlete, Latisha, at six years of age, began advocating for herself. Outside of the program, her mother observed her asking to hold her sister’s and friends’ hands to run on the playground rather than being left behind.

### 4.3. Barriers: “It Was Tricky”

Although the athletes and coaches were excited to be part of this program, there were some clear barriers, which included technology, space and equipment, and age.


**Technology: “We were very limited through a screen”**


Many of the pairs experienced internet issues when trying to connect through Zoom or Facetime for their scheduled lessons. Jose shared, “It was virtual so it was tricky. Um, sometimes, a few times, either I was waiting in the room, it said ‘please wait’ to let you resume, and then it never let me in, so, a little technical difficulties.” The physical education teacher shared, “…getting online independently is a bit of a challenge for some students, so if they didn’t have a parent home or weren’t with someone who could help them, they had to miss the session.”

Nora said “… my siblings are also on online school, so there was there were some Internet issues where I would drop out or I couldn’t hear her or she couldn’t hear me, but that’s about it.” 

More challenging for the coaches and athletes was the challenge of accessible cues and feedback. Many coaches struggled to see their athletes during the lesson because the athlete couldn’t visually see where the camera was facing, and they were often out of visual range. Additionally, many athletes couldn’t see the demonstrations of their coaches on the screen and some coaches struggled to provide descriptive enough cues to accompany the exercises. Brian expressed, “Oh, I noticed that I couldn’t, I didn’t know exactly what they were showing me. It would have been kind of hard on Zoom or Facetime or something to show that.” The practicum instructor heard this time and again from her students, stating, “…if a child was blind, they didn’t know if their camera angle was on them or not, so the coach couldn’t always give them feedback.” Coach Rachel expressed:

…being online and working on a screen, she couldn’t see my demonstration; she wasn’t able to... I wasn’t able to like model physically for her or move her or help adjust or anything; it was all verbal almost, and that doesn’t work all the time… And she would get up to the screen and I would try to do it, but I’d be too far back and if I moved up, she couldn’t see my whole body so it was like just a really tough situation with modeling.

Coach Jordan found ways to overcome the issue: “I found it challenging, but I was able to adjust as time went on and learn how to give the proper cues and things like that.”


**Space and Equipment: “I adapted to the area and just use what he had”**


Several coaches noted that space and equipment were limited for their athletes. Coach Rachel expressed, “He could lay, like his entire body could go on the floor like he could lay down to do a push-up and everything. That’s really the only space he had to do exercises in.” The practicum instructor saw this too:

I think some of the kids didn’t have the proper equipment, like if they wanted to do baseball or they wanted to do basketball with the bell ball, they might not have had that equipment, some of the kids didn’t have the space to go running outside, or to do their fitness outside.

Interestingly, the coaches were the only ones who found the limited space and equipment of the athletes a barrier. This shows how well the coaches were able to adapt their lesson plans to utilize the available space and equipment, so the athletes didn’t feel limited in the activities. Coach Rachel shared, “… all the exercises we did were just body weight because she had really limited space to work with. She only had a tiny bedroom.” Coach Jamal had a similar experience:

…so we did a lot of stationary workouts like different things like push-ups or stretches or we just kind of, I adapted to the area and just use what he had, and the ceiling isn’t very high either, so we couldn’t do things like jump squats or anything that involved him jumping.


**Age: “The whole age thing has a factor”**


The coaches in the program identified the age of the athletes as a barrier. Many coaches who worked with a younger athlete experienced more challenges than those who worked with older athletes. The younger athletes struggled to stay on task without a support person in the home. When the athletes were younger, all communication had to go through the parents. If the parent wasn’t home when immediate communication was needed, the athlete often missed out on the lesson. This was clear with Coach Holly:

I think the whole age thing has a factor, too because Latisha was only six, yeah. So, it was very hard to have her focus on just me when she has like three other siblings too that are running around the house at the same exact time.

Coach Heather had a similar experience:

…it was just a struggle to get her to do anything. So, I enjoyed it a lot because we would start off really well and then the end of the lesson would go downhill, and her mom would just leave her in the room and kind of just leave us alone with her. And we would try to talk to her, and she was just in the zone where she didn’t want to pay attention to us. So, I think it’s really hard. I think maybe working with older kids would have been a bit easier because we had a six year-old so she kind of, I think she struggled with knowing what we had planned for her, and if her mom wasn’t there I don’t know if we would have been able to do as much.

Along the same lines, Coach Marcus expressed:

I think age was clearly a barrier too because Jose’s 10 and Adam is 17, so everything with Jose I would have to go through his mom and I couldn’t just text him hey like could you actually meet at 3:15 instead of 3:30 today, or does this time still work? Everything it was just... there was one time where I had logged on, and he said he had logged on and then no, there was nothing. I waited 30 minutes, and then I was like okay I guess we’re not meeting today, maybe a conflict came up but like he couldn’t communicate that with me, and then the mom emailed me as I’m going to work and was like, oh Jose logged on and you weren’t there. I was like what? I was like, oh okay maybe the Zoom link’s broken because we both were there, so it was just like a couple issues that would happen, because he’s 10, he doesn’t have a phone, you can’t just text him. 

## 5. Summary and Implications

This study illuminated a better understanding of the experiences of youth with visual impairments in virtual practicum programs through listening to their voices. The qualitative nature of the experiences of the athletes and the professional preparation students in this study is an expansion of previous research in this area [21,22,23]. This information is vital to developing effective programming for this population [38]. The purpose of this study was to examine youth with visual impairments and their coaches’ experiences in a three-month virtual physical activity practicum. Three major themes emerged from interviews with the athletes, practicum instructor, and physical education teacher, as well as the focus group interviews with the coaches following the intervention. The themes included: (1) friendship, (2) self-determination, and (3) barriers.

One of the benefits of this program was the long duration of three months. Most practicum programs are 6 to 10 weeks long. This duration is closer to the length of Sato and Haegele [23] and Standal et al. [22]. The long duration of the practicum is also likely to have positively affected the professional preparation students’ level of self-efficacy. Research suggests that the more experience pre-service teachers gain in learning how to instruct students with disabilities, the higher their self-efficacy [39,40].

To the authors’ knowledge, this is the first study to examine the experiences of athletes with visual impairments and their coaches regarding a virtual physical activity practicum. Although this study provides some promising results, further research should be conducted to examine the effectiveness of virtual physical activity programming in youth with visual impairments upon self-advocacy, socialization, and motor competence. Second, it is important to note that a friendship theme developed through this experience, considering there were no in-person sessions. Third, it is important to recognize that a theme of self-determination emerged from this study with three subthemes: autonomy, competence, and self-advocacy. Self-determination and socialization are both components of the Expanded Core Curriculum (ECC) which were also found to improve across a one-week in-person sports camp for youth who are visually impaired [24]. Finally, barriers were revealed from this current experience, including technology, space and equipment, and age. The barriers of technology, age, and space in this study were unique to the current situation. The students were often limited to a living room or bedroom due to COVID-19 restrictions and the inability to go outside due to the internet connectivity. There were no other studies to date that revealed these as barriers per se. The barrier of equipment was found in many articles over the years [41,42] and seems to be a persistent issue when instructing children with visual impairments.

## 6. Limitations

The current study was conducted during the height of the COVID-19 pandemic. The professional preparation students needed practicum experience, and the youth with visual impairments needed physical activity opportunities. This was a convenience sample with limitations in space, technology, internet availability, and time constraints. The time that the youth were available did not always match the time the professional preparation students were available. There was also, at times, a lack of adequate equipment in the homes of the youth. Lastly, children who were too young to have a cell phone at times found it challenging to connect with their coach (professional preparation student) when the parents were unavailable. The strengths of this study were the study duration of three months, the often-enthusiastic engagement of the youth with visual impairments, and the 1:1 nature of the practicum.

## 7. Conclusions

This study illuminated the benefits of a virtual practicum program for children with disabilities. The benefits of the program were tremendous and were related to physical activity improvements, self-determination, socialization, and friendships. The fact that the program was inexpensive was a plus, as this can be a barrier to some programs related to transportation and equipment. In addition, the fact that the youth and the 1:1 coach chose their times to meet was individualized gave them autonomy and self-determination. This study can be replicated with youth with any disability. It can be experienced with the children participating alone or with their parents and siblings. It can also be conducted with more than one professional preparation student, as in a few instances, several of our students Zoomed in with their athlete. Lastly, it can be conducted with a specific curriculum to follow specific skills and activities, or as we did with the program, following the preferences of the athlete and their coach.

## Figures and Tables

**Table 1 ijerph-19-00841-t001:** Demographic information.

Name & State	Age	Sex & Ethnicity	Level of VI	Cause of VI	Number of Previous Camps	Number of Meetings & Focus
Adam NY	17	M Caucasian	B1	Lebers congenital amerosis Birth	25+	58+ 2 in-person Fitness Ice hockey
Karen NY	14	F Caucasian African American	B1	Septo optic dysplasia Birth	6X	6 Fitness
Riad NY	15	M Caucasian	B3	Cone dystrophy Birth	6X	28X 2 in-person Fitness
Nayla NY	17	F Latinx	B3	Prematurity Birth	0	40X Fitness Dance
Latisha GA	6	F African Middle Eastern	B1	Anophthalmia Microphthalmia Birth	0	12x Run Locomotor Language
Jose MA	10	M Caucasian	B3	Albinism Birth	0	6x Fitness basketball
Sam NY	13	M Caucasian	B3	Cone rod dystrophy Birth	1x	6X Soccer Fitness/run Yoga basketball
Tim NY	11	M Asian	B1	Toxicity during pregnancy Birth	2 F2F 3 virtual	10X Jujitsu Fitness

M = Male; F = Female; B1—totally blind; B2—travel vision, 20/600 degree of vision and up; B3— legally blind, 20/200-20/599 degree of vision.

**Table 2 ijerph-19-00841-t002:** Focus Group Demographics.

Name	Age	Gender	Athlete’s Age	Athlete’s Level of Visual Impairment	Athlete’s Gender
Rachel	21	F	13	B2	M
Gary	20	M	11	B3	M
Jordan	20	M	13	B2	M
Heather	20	F	6	B1	F
Jamal	32	M	11	B1	M
Holly	21	F	17	B1	F
Marcus	21	M	15	B2	M
Nick	21	M	17	B1	M
Sylvie	21	F	17	B1	F
Elena	21	F	6	B1	F
Kayla	20	F	10	B3	M

M = Male, F = Female, B1—totally blind, B2—travel vision, 20/600 degree of vision and up; B3— legally blind, 20/200-20/599 degree of vision.

## Data Availability

Not applicable.

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
