# Peer review of "A Qualitative Inquiry of a Three-Month Virtual Practicum Program on Youth with Visual Impairments and Their Coaches"

_ijerph, 2022, doi:10.3390/ijerph19020841_

Round 1
Reviewer 1 Report
First of all, I would like to thank the authors for this paper, which deals with an issue that is both exciting and relevant. The question of teacher preparation with regard to teaching students with disabilities is fundamentally a relevant question in sport pedagogy, where new ways, as proposed and exploratively researched in this article, can be an important enrichment. Against the background of the high relevance of practicum programs in this context and the simultaneously – not only since the Corona pandemic – increasingly important role of digital formats, the article enters significant new territory with the question of whether or how a virtual practicum program can be sensibly and successfully designed. Nevertheless, some weaknesses in the structure of the article, in the justification and presentation of the methodological approach and in the presentation of the results are apparent and should be remedied. These points are outlined below.
- Regarding the semi-structured interviews, the authors outline some guiding questions which illustrate how the interviews were structured. However, the paper does not explain the reasons for the questions. Why are these points asked about and not something else? What Interest is being pursued in detail with these questions? The questions seem plausible, but do not all necessarily arise from the previously excellent argumentation and the excellent presentation of the state of research. Rather, it sometimes appears that the questions arise more or less spontaneously. Here, it should be better justified why exactly these questions are asked in this context. It is helpful that the scripts were send to experts but that doesn't relieve the need to justify the questions. Without this justification, the study threatens to lose some of its depth, which I believe it has the potential to achieve.
- The article focuses and presents experiences in the online practicum program. Especially in the presentation of the results and their discussion, attention should be paid to the fact that several different levels of experience and several different pedagogical processes are relevant here. On the one hand, it is about the experiences of the coaches, on the other hand about those of the students. In the practicum program, coaches are undergoing a professionalization process in which they are somehow learning to teach PE. The students, in turn, also appear in the program as learners. In my opinion, the text occasionally gets a little confused about whose experiences are involved in whose learning, and it is not always entirely clear. In this respect, this presentation of the results could be a bit more structured.
- Some of the barriers found appear to be typical pedagogical barriers that similarly exist in other pedagogical settings. For example, the issue of different ages is certainly not unique to the online practicum setting. Likewise, it is unsurprising that coaches in particular, as the ones who orchestrate the teaching-learning setting, see spatial and equipment-related barriers, rather than students perceiving their familiar environment (their own home) as a barrier. In this respect, it would be helpful here to dock a bit more to educational theory with regard to the interpretation and discussion of the results.
- Overall, a critically reflective discussion of the results is, in my perception, too short in the article. The results would have the potential to produce much more interpretative depth and thus lead to a more differentiated discussion. The article would undoubtedly benefit from this.
- Editorial note: Some misplaced hyphens appear in the text.
Author Response
Thank you for your thoughtful reviews. This feedback has helped to make this article even better. *All changes are in yellow highlight.
Related to your first point. We highlighted the purpose more clearly. We added this for clarity "The lead researchers began with the end in mind by developing questions that would provide the answers to critical information needed to understand the effectiveness of the program. The overall purpose of the questions was to understand the lived experience of the virtual program by the athletes and the coaches"
Related to your 2nd point, this explanation was added to clarify the presentation of the findings and discussion. "The results highlight the effects of the program on both the athletes learning of skills and improved physical activity, and the coaches understanding of pedagogical instructional approaches. Please note that these themes and sub-themes in the findings reflect the experiences of both the athletes and the coaches as they worked together and acquired these experiences together. Clarification was made in each quote if the response was from an athlete or a coach."
Your last point was a good point. We have highlighted where we state that the findings are similar to previous studies. These barriers were also found to be unique to this particular situation as highlighted in the 2nd to last paragraph of the Summary and Implications section. This addition helps clarify the similarities and differences from this study. This certainly gives us some ideas for further research and analysis in the future with similar programming.
Reviewer 2 Report
It's interesting to discuss the experiences of students with visual impairments (VI) or preservice physical education (PE) teachers in some kind of practicum program. However, this manuscript was lack of academic logic and writing. Please see the following comments:
- Introduction Part.
- The purposes of the study were unclear. The manuscript was talking about the effects of a three-month virtual practicum program? Or talk about the attitude of the students with VI and PE teachers towards this program? Can not see the specific research problem. By the way, the preservice PE teachers and the coaches are two different concepts. Why the authors use these two different concepts to discuss one role?
- The manuscript to discuss many studies which examine the effects of some programs on the confidence, or self-efficacy, or attitudes of preservice PE teachers. These were not related to the theme of the study. Too many variables. The paper aimed to examine the effects of the programs on the characteristics of the preservice PE teachers?
- The Method Part
- The manuscript did not present the principles of sample selection.
- The most important thing in the qualitative study method part is how to develop the interview or focus group questions. The manuscript did not present it.
Duo to the purposes of the study was very unclear, the results of the study is unreliable.
Author Response
The study looked at the effect of the practicum program on the pre-service teachers and the students. The goal of the program was to improve the skills of the children and to improve the teaching experience of the teachers. The results looked at both groups as they worked together in the program. The goal was to analyze the effect of the program on both groups of participants. This was clarified in the paper. (See highlighted sections)
The introduction did discuss the many benefits of practicum programs. This set the stage for the importance of the study. We did not do any pre and post program assessments on the practicum students. This was not an aim of the study. We aimed to find out the lived experience of the athletes and pre-service teachers. We did not say that we were measuring self-efficacy or any other pre-service teacher assessment. This was just making the case for the study. These types of practicums often improve many aspects of teaching for the pre-service teachers.
We clarified the purpose of the program and the purpose of the study. Please note that this is a qualitative descriptive study of the experiences of the athletes and their coaches (pre-service teachers) after a three-month virtual program.
The sample selection of the participants was more clearly described in the methods. See the yellow highlights. Thank you
The purpose and the description of the development of the questions was also addressed.
Reviewer 3 Report
This paper is very interesting. Thank you for the opportunity to review this interesting article. It surely contributes to a better knowledge of assess the experiences of youth with visual impairments. ​I think this study is sound and should be published. However, some parts of the paper should be improved.
- Abstract: The summary should follow the style of structured summaries (background, methods, results, and conclusions but no heading). It should describe the objective and briefly, the main methods or treatments applied, as well as the number of participants in the study.
- The general objective and specific objectives should appear at the end of the introduction. The objective should be clearly written, referring to the population, the intervention, the comparison and the results (PICO strategy).
- Did you realize a sample size estimation? If not, why?
- The sample is very small for this type of study that should pay careful attention to its inference results, and should be limited in the article.
Author Response
Thank you very much for your review of our manuscript. I am not sure if you had the chance to read the entire article itself.
- Our abstract was formatted according to the manuscript guidelines.
- We added the objectives of the study in the abstract. Thank you.
- This study was not a quantitative study so no sample size estimation is necessary. This was a phenomenological study so no sample size estimation was necessary.
- The sample size in this qualitative study is actually very robust related to typical sample sizes in qualitative research (21 participants). We do not address this as it is not an issue.
Round 2
Reviewer 2 Report
I will keep my original opinion on this paper. No matter the quantitative research or qualitative research, the purposes of the study should be focused on one group participants.
Author Response
The purpose of our study was to determine the experiences of the participants of a three-month virtual practicum experience. There have been no studies to date that analyzed this approach. To single out only the coaches or the participants will only provide half of the findings. It is extremely important to not only determine the influence this has on the professional preparation students, but also the impact it has on the children that it endeavors to support. This is a qualitative study with two full sets of data: interviews and focus groups. The program is fully interconnected and to take away the voice of one of the populations affected is diminishing the study and taking away the full honest true findings. The total picture of the effect of the program would be lost.